# Guiding 3D U-nets with signed distance fields for creating 3D models from images

**Kristine Aavild Juhl**                                    KAJUL@DTU.DK
**Rasmus Reinhold Paulsen**                                RAPA@DTU.DK
**Anders Bjorholm Dahl**                                    ABDA@DTU.DK
**Vedrana Andersen Dahl**                                  VAND@DTU.DK
*Technical University of Denmark, Kongens Lyngby, Denmark*

**Ole De Backer**                                OLE.DE.BACKER@REGIONH.DK
**Klaus Fuglsang Kofoed**                        KLAUS.KOFOED@REGIONH.DK
*Department of Cardiology, Righospitalet, University of Copenhagen, Copenhagen, Denmark*

**Oscar Camara**                                    OSCAR.CAMARA@UPF.EDU
*Universitat Pompeu Fabra, Barcelona, Spain*

## Abstract

Morphological analysis of the left atrial appendage is an important tool to assess risk of ischemic stroke. Most deep learning approaches for 3D segmentation is guided by binary labelmaps, which results in voxelized segmentations unsuitable for morphological analysis. We propose to use signed distance fields to guide a deep network towards morphologically consistent 3D models. The proposed strategy is evaluated on a synthetic dataset of simple geometries, as well as a set of cardiac computed tomography images containing the left atrial appendage. The proposed method produces smooth surfaces with a closer resemblance to the true surface in terms of segmentation overlap and surface distance.

**Keywords:** Signed distance fields, pixel-wise regression, left atrial appendage

## 1. Introduction

The left atrial appendage (LAA) is a complex tubular structure originating from the left atrium (LA). The LAA is known to have large morphological variabilities between patients and studies have shown a correlation between the LAA morphology and the risk of ischemic stroke (Di Biase et al., 2012). This makes morphological analysis of the LAA a relevant topic. Cardiac computed tomography angiography (CCTA) images of patients with suspected risk of thrombus formation in the LAA can be used for generating high resolution 3D models of the LA and LAA useful for morphological analysis. Manual segmentation of the CCTA images is possible but not viable for large population-based studies. Automatic deep learning methods can be a good alternative, where especially the U-net architecture has proven powerful (Ronneberger et al., 2015). However, the traditional 2D U-net cannot capture the 3D consistency in the image, whereas the 3D counterpart is limited to work on low resolution patches due to memory restrictions. Despite this, the 3D U-net shows good results in terms of Dice-score in many applications, but excessive voxelization of the 3D segmentation makes it unsuitable for morphological analysis.

In this paper we propose to use signed distance fields (SDFs) to guide a 3D U-net towards creating morphologically consistent 3D models of anatomical structures. The SDF is more suitable for representing a smooth surface than a binary volume and provides additional 3D topological information to the network without increasing the computational burden.

## 2. Methods

To compare the effect of representing the anatomical shape as a SDF instead of a binary occupancy grid, two similar networks are used. The pixel-wise classification (PWC) network is a standard 3D U-net predicting binary labelmaps. The input image is downsampled to $64^3$ using linear interpolation and the $64^3$ output labelmap is upsampled to the original resolution using nearest neighbour upsampling. The network is trained with a cross-entropy loss.

Predicting a SDF is no longer a classification task but a regression problem. The pixel-wise regression (PWR) network is kept as similar to the PWC architecture as possible, to ensure the same learning capacity. To optimize the network for the regression task, the softmax activation of the final layer is replaced by a linear activation function, as it was seen in other PWR networks (Yao et al., 2018). Both downsampling of the input image and upsampling of the output SDF is done using linear interpolation. The used loss function is a mean squared error (MSE) inversely weighted with the absolute value of the SDF. This ensures the network prioritizes the learning around the contour of the anatomy.

The performance of the two methods is evaluated based on volume overlap and surface distance. The volume overlap is measured as Dice-score for the entire segmentation and a contour-Dice, where the Dice-score is calculated only inside a mask around the segmentation contour. The mask is created from dilation and erosion of the true segmentation with a $5 \times 5 \times 5$ kernel. The surface distance is measured as an average symmetric distance (ASD) and root mean square symmetric distance (RMSD).

## 3. Experimental results

**Synthetic data: Geometric shapes**   A synthetic dataset is created with cuboids, rhomboids, ellipsoids and cylinders of different sizes and rotations in a $512^3$ binary grid. The two networks are trained on 19 examples of each shape (76 examples in total) and tested on 6 examples of each shape (24 examples in total). The evaluation results and examples of predictions using the two methods together with the true surface are seen in Table 1 and Figure 1 respectively.

**Medical data: Segmentation of left atrium including left atrial appendage**   The method is further evaluated on a medical dataset consisting of 30 CCTA scans with manually annotated LA and LAA. The CT images are acquired by the Department of Radiology, Rigshospitalet, University of Copenhagen. The acquired CT volumes are of $512 \times 512 \times 560$ voxels with a voxel size of $0.5 \times 0.5 \times 0.25$ mm. A region of interest is extracted around the LA and LAA resulting in average ROI size of $375 \times 375 \times 405$ voxels. The PWR and PWC networks are trained on 25 images and tested on 5 images. The evaluation results are seen in Table 1 and Figure 1 shows the surface from the manual annotation together with examples of outputs from the two methods.

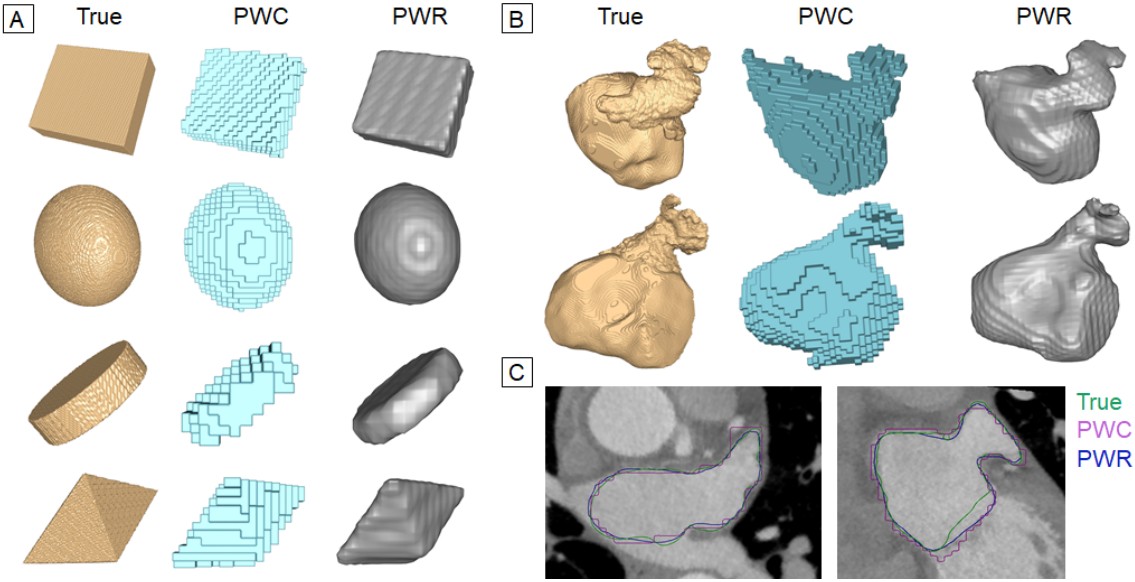

Figure 1: A: Results on selected test examples from synthetic dataset. B: Results on selected test examples from medical dataset. C: Contours on axial and coronal slice of one example. PWC: Pixel-wise classification, PWR: Pixel-wise regression.

Table 1: Average evaluation metrics on test images from the synthetic and medical dataset.

|  | Synthetic | | | Medical | | |
|---|---|---|---|---|---|---|
|  | **PWC** | **PWR** | **Gain** | **PWC** | **PWR** | **Gain** |
| Dice(×100) | 92.38 | 97.08 | 5.09% | 89.97 | 92.04 | 2.30% |
| contourDice(×100) | 68.49 | 87.69 | 28.03% | 63.88 | 72.18 | 12.99% |
| ASD | 1.573 | 0.714 | 54.61% | 1.267 | 1.097 | 13.41% |
| RMSD | 1.947 | 1.018 | 47.71% | 2.087 | 1.695 | 18.78% |

## 4. Discussion and conclusion

Based on the results in both experiments, it is evident that the PWR method creates surfaces that on average are closer to that of the actual anatomy and these surfaces look more realistic due to the smooth nature of the normals. Guiding the U-net with SDFs results in an increase in overall Dice-score, where the largest improvement is seen around the contour. While the gross-structure of the LA and LAA is preserved in the predicted surfaces, the high frequency details are lost due to the low resolution sampling of the SDF.

For future work we plan to evaluate how well these low-frequency shapes preserve important morphological and clinical parameters such as LA diameter, LAA length, LAA orifice diameter, curvature, etc. Furthermore the method is to be evaluated on larger medical databases, where 3D models of anatomical structures also are of interest.

## Acknowledgments

This work was supported by the Spanish Ministry of Economy and Competitiveness under the Maria de Maeztu Units of Excellence Programme (MDM-2015-0502) and the Retos I+D Programme (DPI2015-71640-R).

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
