# OpenReview forum: "Guiding 3D U-nets with signed distance fields for creating 3D models from images"
_MIDL.io/2019/Conference/Abstract — MIDL Abstract 2019_

### Official Review · AnonReviewer2 · 2019-04-29

**Rating:** 3
**Confidence:** 3

**Review:**

The interesting part in this work is to use signed distance fields to seek morphologically consistent 3D models. Synthetic data and real data demonstrated promising results.

---

### Official Review · AnonReviewer1 · 2019-05-02
**New approach for training segmentation but problems with interpolation**

**Rating:** 3
**Confidence:** 2

**Review:**

Pros:
- A new approach to generating smoothing segmentations when training at lower resolution is required.
- Weighting with the signed distance field should help the model concentrate on learning the contour of desired segmentation.
- Writing/figures are clear.

Cons:
- The model is trained to estimate a signed distance field, but there is nothing in the model that will in fact constrain the resulting prediction to be an actual signed distance field.
- The biggest issue I see is that the comparison to the binary labelmap approach is unfair and not what I would consider standard practice. The authors upsample the output low resolution labelmap with nearest neighbors. I think standard practice that would produce greatly improved segmentation results is to upsample the output probability map from the softmax activation layer (with likely linear interpolation), then extract the binary map at the original resolution. In the future the authors should use this approach or something similar as comparison rather than interpolation of the binary map.
- It seems that the values in the last 2 columns of Table 1 need to be switched (PWR and Gain).

---

### Decision · Program_Chairs · 2019-05-06
**Acceptance Decision**

Accept